# Modulation of Cathepsin S (*CTSS*) Regulates the Secretion of Progesterone and Estradiol, Proliferation, and Apoptosis of Ovarian Granulosa Cells in Rabbits

**DOI:** 10.3390/ani11061770

**Published:** 2021-06-13

**Authors:** Guohua Song, Yixuan Jiang, Yaling Wang, Mingkun Song, Xuanmin Niu, Huifen Xu, Ming Li

**Affiliations:** 1College of Animal Science and Technology, Henan Agricultural University, Zhengzhou 450046, China; m13782525956@163.com (G.S.); J19203090@163.com (Y.J.); 15637410238@163.com (Y.W.); m18625951565@163.com (M.S.); 2Yangguang Rabbit Technology Co., Ltd., Jiyuan 454650, China; damoguyue@163.com

**Keywords:** CTSS, granulosa cells, proliferation, apoptosis, hormone secretion

## Abstract

**Simple Summary:**

In goat and sheep, *CTSS* is reported to be important for the development and maturation of oocytes by regulating cell proliferation and apoptosis. The purpose of this study was to investigate the role of *CTSS* in regulating cell apoptosis and hormone secretion in rabbit granulosa cells. Our results suggested that the *CTSS* gene can promote the proliferation of granulosa cells and reduce its apoptosis in vitro, while overexpression of *CTSS* promoted the secretion of progesterone and estrogen in rabbit granulosa cells. Therefore, manipulation of *CTSS* may improve development of oocytes, and thus provide an approach for better manipulation of rabbit reproductive performance.

**Abstract:**

Cathepsin S (*CTSS*) is a member of cysteine protease family. Although many studies have demonstrated the vital role of *CTSS* in many physiological and pathological processes including tumor growth, angiogenesis and metastasis, the function of *CTSS* in the development of rabbit granulosa cells (GCS) remains unknown. To address this question, we isolated rabbit GCS and explored the regulatory function of the *CTSS* gene in cell proliferation and apoptosis. *CTSS* overexpression significantly promoted the secretion of progesterone (P4) and estrogen (E2) by increasing the expression of *STAR* and *CYP19A1* (*p* < 0.05). We also found that overexpression of *CTSS* increased GCS proliferation by up-regulating the expression of proliferation related gene (*PCNA*) and anti-apoptotic gene (*BCL2*). Cell apoptosis was markedly decreased by *CTSS* activation (*p* < 0.05). In contrast, *CTSS* knockdown significantly decreased the secretion of P4 and E2 and the proliferation of rabbit GCS, while increasing the apoptosis of rabbit GCS. Taken together, our results highlight the important role of *CTSS* in regulating hormone secretion, cell proliferation, and apoptosis in rabbit GCS. These results might provide a basis for better understanding the molecular mechanism of rabbit reproduction.

## 1. Introduction

Rabbits are multi-purpose domestic animals, which can be used as pet animals or biomedical model animals for scientific research, but in people’s lives, rabbits are mainly used for meat and fur production [1,2,3]. Rabbit meat has high nutritional value and is widely accepted by many people for its low content in fat, cholesterol, and high content in protein [4,5]. However, as the demand for rabbit meat increases rapidly, global rabbit meat production is still relatively low [6]. Accelerating the reproductive performance is an important strategy to increase rabbit meat production, thus it is vital to investigate the regulatory mechanism of factors affecting rabbit reproduction.

Ovarian granulosa cells (GCS) are an important component of follicles and play a vital role in follicle growth and development. Each follicle contains an oocyte, numerous GCS and theca cells [7]. GCS deliver regulatory signals to oocytes through zone projections (TZPs), and also provide nutrition and metabolite for oocyte [8]. Many studies reported that apoptosis of GCS affect cellular connection between GCS and oocytes [9]. It causes ovarian follicular atresia [10], which occurs in more than 99% of developmental follicles in mammals [11,12]. Another important role of GCS is the secretion of progesterone and estradiol, both of which play an important role in animal breeding and reproduction. It was reported that estradiol can promote the expression of FSH receptors in GCS [13], and promote the formation of rat antral follicles [14]. In bovine, low concentration of progesterone enhanced the follicular development, and the diameter of follicular increased [15,16]. Moreover, after follicle ovulation, GCS differentiated into luteal granulosa cells to form the corpus luteum. The main function of corpus luteum is to secrete progesterone to maintain pregnancy. Therefore, both GCS progesterone secretion disorder and GCS apoptosis will result in a decrease in rabbit reproduction performance.

Cathepsin S (*CTSS*), which is located on chromosome 13 in rabbits, is one member of 11 cysteine proteases, and plays an important role in extracellular matrix degradation and remodeling, antigen presentation, inflammation, and angiogenesis [17]. In malignant tumors, *CTSS* induces tumor proliferation, invasion and metastasis through various mechanisms [18,19,20]. It was reported that *CTSS* might be a potential predictor of chemotherapy response because up-regulation of *CTSS* is associated with tumor progression and poor prognosis [21]. Studies in vitro also showed that *CTSS* promoted adipocyte differentiation by degrading fibrinolytic proteins [22]. Studies have also found that the polymorphisms of the *CTSS* gene are associated with obesity-related traits [23], and *CTSS* circulating levels are associated with triglycerides synthesis and accumulation [24,25]. *CTSS* can also regulate blood sugar by reducing glucose output [26].

The regulatory function of *CTSS* in cell proliferation has been widely investigated, while its function in the development and maturation of rabbits’ oocytes has not been reported. Our hypothesis was that *CTSS* plays an important role in follicle development and ovulation through regulating cell proliferation and hormone secretion. To verify this hypothesis, the present study aimed to investigate the functional role of the *CTSS* gene in cell proliferation and apoptosis, as well as progesterone and estradiol secretion in rabbit GCS. These results provide evidence with which to further uncover the regulatory mechanism which mediated the ovulation process and reproductive performance of rabbits.

## 2. Materials and Methods

### 2.1. Ethic Statement

The present study was designed and performed according to the guidelines of Institutional Animals Care and Use Committee College (IACUC) of College of Animal Science and Technology of Henan Agricultural University, China (Permit Number: 11-0085; Data: 06-2011).

### 2.2. Tissue Sample Collection

In this experiment, three female New Zealand white rabbits (180 days old) were selected from the animal house of Henan Agricultural University. The rabbits were anesthetized and slaughtered for ovary collection. The collected ovaries were placed in PBS (containing 1% penicillin/streptomycin, 37 °C) for rabbit granulosa cells isolation.

### 2.3. Isolation and Culture of Rabbits GCS

The rabbits’ GCS were isolated and cultured according to the previously published papers with few modifications. Briefly, the collected ovaries were washed three times with PBS (phosphate buffer saline) supplemented with 1% penicillin/streptomycin at 37 °C. Follicles were needled with a 1 mL syringe in a basal medium supplemented with DMEM/F12 medium (Gibco, MD, USA), 15% FBS (Gibco, CA, USA), and 1% penicillin/streptomycin (Hyclone, Logan, UT, USA) [27]. The GCS were centrifuged (5 min, 1000 rpm/min) and incubated with basal medium at 37 °C in 5% CO_2_; cell medium was changed every 24 h [28]. The GCS isolated from the 3 rabbits were isolated individually and pooled together for experimental treatments. Before experiments, GCS were seeded in 6-well plate at a density of 3 × 10^5^ cells per cm^2^. After 24 h incubation (approximately 90% confluence), GCS were treated with siRNA and recombinant adenovirus for 24 h. All treatments were performed with 3 independent biological replicates.

### 2.4. Immunostaining

FSHR (follicle stimulating hormone receptor) protein is usually used as a marker protein to identify GCS. The cells were seeded on culture plates plated with cell-climbing slices. After reaching 70% confluence, cells were fixed with 4% paraformaldehyde and then permeated with 0.1% Triton X-100. For immunohistochemical staining, the cells were incubated with anti-FSHR (MAB65591, 1: 200, R&D Systems, Minneapolis, MN, USA) for 1 h at 37 °C, and then incubated with goat anti-rabbit IgG (bs-0296G-FITC, 1:400, Bioss, Beijing, China) at 37 °C for 45 min. Finally, the cells were taken out and mounted with DAPI (4′,6-Diamidino-2-phenylindole, Dihydrochloride) medium (Solarbio, Beijing, China). The cells were then observed and imaged under a fluorescence microscope (Nikon Eclipse C1, Tokyo, Japan).

### 2.5. HE (Hematoxylin-Eosin) Staining

The morphology examination of GCS was measured by HE staining. GCS were seeded in 6-well culture plates plated with cell-climbing slices. Then, the cells were fixed with 95% ethanol for 20 min and stained with hematoxylin dye solution for 3 min. After dehydration with gradient alcohol, the cells were dyed with eosin dye solution for 5 min. Finally, the cells were mounted with neutral gum and observed by microscope (Nikon Eclipse E100, Tokyo, Japan).

### 2.6. SiRNA (Small Interfering RNA) Interference

GCS were seeded into 6-well plates and transfected with siRNA-*CTSS* and siRNA-NC by using Lipofectamine 3000 (Invitrogen, Carlsbad, CA, USA) according to the manufacturer’s instructions. The specific siRNA sequence targeting *CTSS* was designed and synthesized by GenePharma (Shanghai, China). siRNA-*CTSS*, sense: 5′-GGAAGAAAGCCUACGGCAATT-3′; antisense:5′-UUGCCGUAGGCUUUCUUCCTT-3′. siRNA-NC, sense: 5′-UUCUCCGAACGUGUCACGUTT-3′, antisense: 5′-ACGUGACACGUUCGGAGAATT-3′.

### 2.7. Recombinant Adenoviruses Generation

The production of recombinant adenovirus has been previously described [29]. In short, *CTSS* was amplified and subcloned into the shuttle vector pAdTrack-CMV. Then the linearized product of pAdTrack-CMV-*CTSS* was transformed into Escherichia coli BJ5183 competent cells containing the backbone vector pAdEasy-1 to obtain the positive recombinants of pAd-*CTSS*. Then, linearized pAd-*CTSS* fragment was transfected into 293 cells for adenovirus packaging and amplification. GCS were infected with Ad-*CTSS* at MOI (multiplicity of infection) value of 40.

### 2.8. RNA Extraction and RT-qPCR

Total RNA of rabbit GCS were extracted by using Trizol reagent (Invitrogen, CA, USA) according to the instructions of the manufacturer. Total RNA was quantified with a Nanodrop ND-2000 spectrophotometer (Thermo Fisher Scientific, Waltham, MA, USA). cDNA was synthesized by using Prime Script RT reagent Kit (Takara, Tokyo, Japan) following the instructions. The SYBR Premix Ex Taq II kit (Takara, Tokyo, Japan) was used for RT-qPCR. Primers used for RT-qPCR were listed in Table 1. *β-actin* was selected as reference gene.

### 2.9. Measurement of Progesterone and Estradiol Secretion

After 48 h of treatment with Ad-*CTSS* and siRNA-*CTSS*, cell culture medium was collected for determination of progesterone and estradiol concentration. Secretion levels of progesterone and estradiol were evaluated with enzyme linked immunosorbent assay (ELISA) according to the instructions of the manufacturer. The kits were all obtained from Nanjing Ruixin Biology (Quanzhou, China). The absorbance was measured using a microplate reader (Bio-Rad) at 450 nm.

### 2.10. Cell Viability Assay

GCS were seeded onto 96-well plates at 1 × 10^4^/well and treated with recombinant adenovirus and siRNA for 0 h, 12 h, 24 h, 36 h and 48 h, separately. Cell proliferation was detected with Cell Counting Kit-8 (CCK8) (US Everbright^®^ Inc., Nanjing, China). According to the instructions of the manufacturer, the cells were added to 10 μL of CCK8 solution and incubated at 37 °C for 2 h. Then, the optical density (OD) value at a wavelength of 450 nm was detected by a microplate reader (Bio-Rad, Hercules, CA, USA).

### 2.11. Cell Apoptosis Analysis

After treatment with recombinant adenovirus and siRNA for 48 h, GCS apoptosis was detected by Annexin V-Alexa Fluor 647/7-AAD Apoptosis Kit (4Abiotech., Beijing, China). According to the instructions of the manufacturer, GCS were digested with trypsin (without EDTA) and resuspended with cold PBS, and then mixed with 5 μL Annexin V/Alexa Fluor 647 at room temperature for 5 min. Finally, the cells were treated with 10 μL 7-AAD and 400 μL PBS. The apoptosis rate of the GCS was measured with a CytoFLEX flow cytometer (Beckman CytoFlex, CA, USA).

### 2.12. Statistical Analysis

All data analyses were conducted using SPSS 22.0 software (SPSS, Chicago, IL, USA). All the treatments were performed in three independent biological replicates and three technical replicates. Statistical analyses were performed with one-way ANOVA to compare the effects of *CTSS* overexpression and knockdown relative to negative control. The fixed effect was treatment and the random effect was replicated in the statistic model. Relative mRNA expression of RT-qPCR was calculated by using 2^−ΔΔCt^ method, where Ct is the cycle threshold. *p* value < 0.05 was indicative of a statistically significant difference, and data are presented as means ± SE.

## 3. Results

### 3.1. Isolation and Identification of Rabbit GCS

Rabbit GCS were identified by HE staining and immunofluorescence by using FSHR antibody, our HE staining results showed that the nucleus of GCS is blue and the cytoplasm is red, and the pseudopodia between the cells was closely connected (Figure 1A). As shown in Figure 1B, cell nuclei were stained with DAPI and presented to be blue; FSHR protein was green and located in the cytoplasm. The positive rate of FSHR was 99%, indicating that the isolated GCS were 99%.

### 3.2. Efficiency of CTSS Overexpression and SiRNA Interference

RT-qPCR was performed to determine the efficiency of Ad-*CTSS* and siRNA-*CTSS* in rabbit GCS. Our results showed that *CTSS* expression was remarkably increased by Ad-*CTSS* compared with Ad-GFP (Figure 2A, *p* < 0.05), and its expression was significantly decreased by siRNA-*CTSS* relative to siRNA-NC (Figure 2B, *p* < 0.05).

### 3.3. CTSS Promotes the Secretion of Progesterone and Estradiol in Rabbit GCS

To explore the role of *CTSS* on rabbit reproduction, we measured the effects of *CTSS* overexpression and knockdown on hormone secretion in rabbit GCS. As shown in Figure 3A,D, genes responsible for steroid hormone synthesis (Steroidogenic acute regulatory protein, *STAR*) and estrogen synthesis (Cytochrome p450 family 19 subfamily A member 1, *CYP19A1*) were remarkably increased and decreased by *CTSS* overexpression and knockdown, respectively (*p* < 0.05). Consistently, levels of progesterone and estradiol in cell medium were significantly increased by *CTSS* overexpression (Figure 3B,C). While *CTSS* knockdown only decreased progesterone secretion (Figure 3E, *p* < 0.05), estradiol secretion was not changed by siRNA-*CTSS* (Figure 3F, *p* > 0.05).

### 3.4. CTSS Promotes Cell Proliferation Activity in Rabbit GCS

After treatment with Ad-*CTSS* and siRNA-*CTSS* for 0 h, 12 h, 24 h, 36 h and 48 h, the proliferation activity of GCS was measured. As shown in Figure 4A, genes responsible for cell proliferation (*PCNA*, proliferating cell nuclear antigen) were significantly increased by *CTSS* overexpression (*p* < 0.05), and cell proliferation activity of the Ad-*CTSS* infected group was remarkably higher than the Ad-GFP infected group (Figure 4B, *p* < 0.05). *CTSS* knockdown decreased *Notch2* expression and cell proliferation activity in rabbit GCS (Figure 4C,D, *p* < 0.05).

### 3.5. Activation of CTSS Deceased Cell Apoptosis in Rabbit GCS

As shown in Figure 5A, compared with the Ad-GFP infected group, mRNA level of *BCL2* (one of the main anti-apoptotic gene) was significantly increased by *CTSS* overexpression (*p* < 0.05). The other important pro-apoptotic gene, *BAX,* was decreased significantly by *CTSS* knockdown (Figure 5E, *p* < 0.05). Flow cytometry analysis revealed that *CTSS* overexpression inhibited the apoptosis of rabbit GCS significantly (Figure 5B–D, *p* < 0.01). As shown in Figure 5F–H, *CTSS* knockdown led to a remarkable increase in apoptotic cells in rabbit GCS (*p* < 0.05).

## 4. Discussions

Reproductive performance is an important index for animal husbandry. GCS play an important role in oocyte development and affect female breeding performance. A number of studies have shown that small molecules including Vitamin D3 [30], FSH [31], Bisphenol A [32] and genes including *CITED4* [33], *CXCL12* [34], and *STAT4* [35] regulate cell proliferation, apoptosis and hormone secretion in mammals, thus affecting estrus of female animals and oocyte maturation. However, as one species of the high fertility husbandry animals, the effect of *CTSS* gene on rabbit reproduction has been rarely studied.

Cathepsin S (*CTSS*), one kind of lysosomal protease, is a major participant in proteolysis [36], and participates in many physiological processes. Studies have shown that *CTSS* gene played an important role in lipid metabolism in tumor tissue. To further investigate the role of *CTSS* in rabbit reproduction, we isolated rabbit GCS. GCS exist in the follicle and play an important role in the development of the follicle, as it can provide nutrients to oocytes during follicular development [37]. Studies have reported many methods for in vitro isolation and culture of GCS in pigs [38], cattle [39], and mice [40]. Isolation of rabbit GCS in the present study was performed according to the published methods with little modifications [41], providing material for further study of the complex mechanism of rabbit reproduction.

Progesterone and estradiol are mainly secreted by GCS and play an important role in the breeding process of mammals. They can promote estrus of female rabbits and early fetal pregnancy. To investigate whether *CTSS* affects progesterone and estradiol secretion in GCS, we overexpressed and interfered the expression of *CTSS* gene in rabbit GCS. Our results demonstrated that *CTSS* gene can regulate the secretion of progesterone hormone, and that this effect might be mediated by *STAR* and *CYP19A1*. As summarized in Figure 6, activation of *CTSS* resulted in up-regulation of progesterone hormone-related genes (*STAR* and *CYP19A1*) and promoted the secretion of progesterone. In mammals, progesterone plays a key role in follicular development and early fetal pregnancy [40]. Progesterone promotes follicular development, and low levels of progesterone increase follicular diameter [16]. However, the insufficiency of progesterone secretion will cause the fetus to be unable to survive or, even worse, will result in the abortion phenomenon [42]. In this experiment, our results demonstrated that *CTSS* play an important role in regulating the secretion of progesterone and estradiol in GCS.

By providing nutrition and metabolite to oocytes, GCS play an important role in oocyte development. GCS also transmit signals and information for oocytes, and make oocytes mature. Apoptosis of GCS affects follicle development and leads to follicular atresia. In this study, we found that *CTSS* gene could affect the proliferation and apoptosis of GCS, and that the overexpression of *CTSS* gene in GCS increased the expression of proliferation gene (*PCNA*) and anti-apoptotic gene (*BCL2*), and promoted the proliferation of GCS while decreasing its apoptosis (Figure 6). Consistently, *CTSS* knockdown decreased the proliferation rate while increasing the apoptosis of rabbit GCS. These findings are in accordance with recent research reporting that *CTSS* interference inhibited the proliferation while inducing the apoptosis in mhCC97-H cells [43,44], and recent studies have also shown that *CTSS* plays a crucial role in the invasion and apoptosis of cancer cells [45,46]. Taken together, our results have demonstrated that *CTSS* can promote the proliferation while reducing the apoptosis of rabbit GCS.

## 5. Conclusions

In summary, our data suggest that *CTSS* regulates the proliferation and apoptosis of granulosa cells by regulating the expression of key genes involved in these processes. In addition, *CTSS* is also involved in the process of progesterone and estradiol secretion in rabbit GCS. Overall, the present study provided evidence that *CTSS* may play an important role in regulating follicular development by regulating cell apoptosis and hormone secretion in rabbits. However, whether manipulation of *CTSS* expression in vivo can improve the reproductive performance of rabbits needs further research.

## Figures and Tables

**Figure 1 animals-11-01770-f001:**
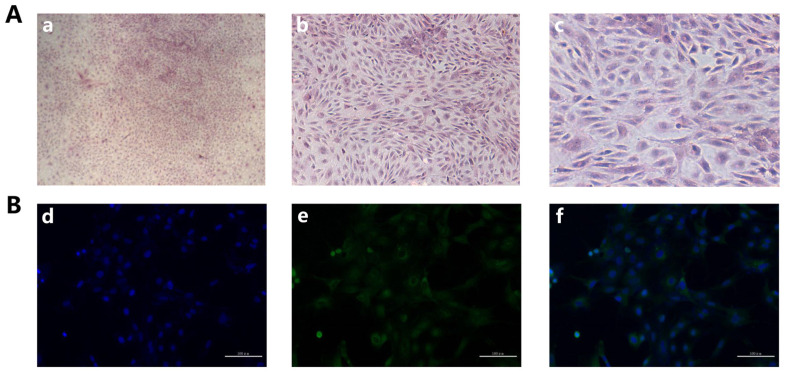
Isolation and identification of rabbit GCS: (**A**): GCS were stained with Hematoxylin and eosin (H&E) staining, (**a**–**c**): Original magnification, 40×, 100×, 200×. (**B**): The identification of rabbit GCS was measured by immunofluorescence staining with fluorescein isothiocyanate (FITC)-goat anti-rabbit IgG. (**d**): Cell nucleus were stained by DAPI (blue), (**e**): FSHR were stained with fluorescein isothiocyanate (FITC)-goat anti-rabbit IgG (green), (**f**): Merge of (**d**) and (**e**). (Scale bar = 100 μm).

**Figure 2 animals-11-01770-f002:**
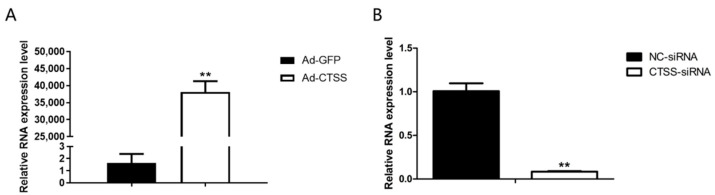
The efficiency of Ad-*CTSS* and siRNA-*CTSS* in rabbit GCS: (**A**): mRNA expression changes of *CTSS* after infection with Ad-GFP. (**B**): mRNA expression changes of *CTSS* after transfection with siRNA-*CTSS*. Values are presented as means ± SEM for three biological replicates. ** *p* < 0.01.

**Figure 3 animals-11-01770-f003:**
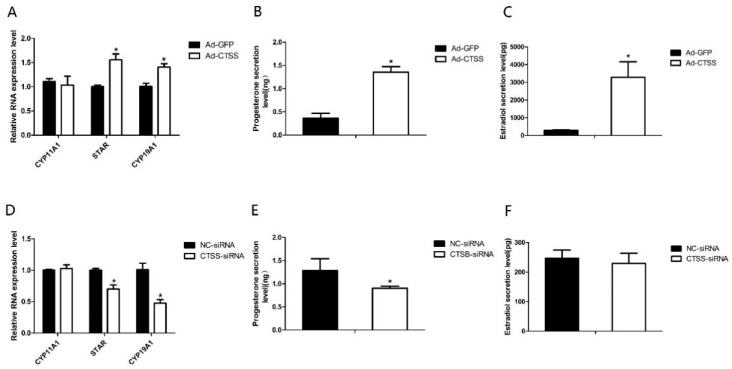
*CTSS* overexpression and interference on progesterone and estradiol secretion in rabbit GCS: Effects of *CTSS* overexpression and knockdown on (**A**,**D**) mRNA expression changes of genes related to progesterone and estradiol secretion; (**B**,**E**) progesterone secretion and (**C**,**F**) estradiol secretion. Values are presented as means ± SEM for 3 biological replicates. * *p* < 0.05.

**Figure 4 animals-11-01770-f004:**
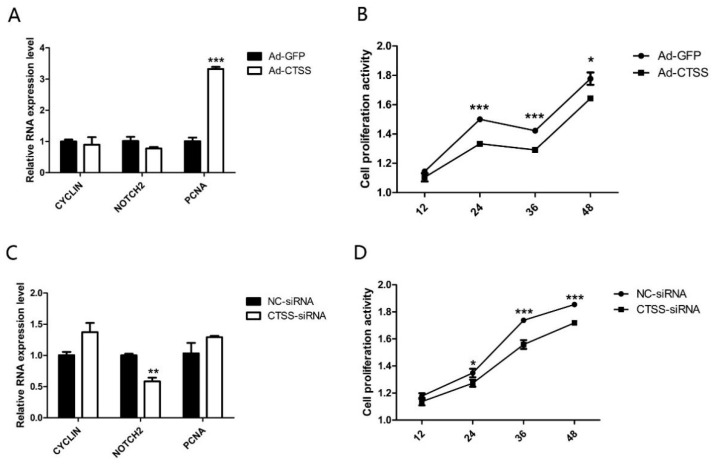
*CTSS* overexpression and interference on cell proliferation in rabbit GCS: Effects of *CTSS* overexpression and knockdown on (**A**,**C**): mRNA expression changes of cell proliferation related genes and (**B**,**D**): cell proliferation activity. Values are presented as means ± SEM for 3 biological replicates. * *p* < 0.05; ** *p* < 0.01; *** *p* < 0.001.

**Figure 5 animals-11-01770-f005:**
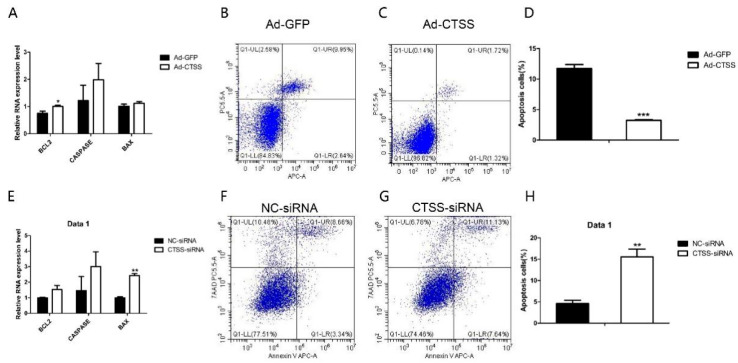
Effect of *CTSS* overexpression and interference on cell apoptosis in rabbit GCS: Relative mRNA expression changes of marker genes related to cell apoptosis after *CTSS* overexpression (**A**) and knockdown (**E**). Cell apoptosis rate was measured by using flow cytometry after *CTSS* overexpression (**B**–**D**) and knockdown (**F**–**H**). * *p* < 0.05; ** *p* < 0.01; *** *p* < 0.001.

**Figure 6 animals-11-01770-f006:**
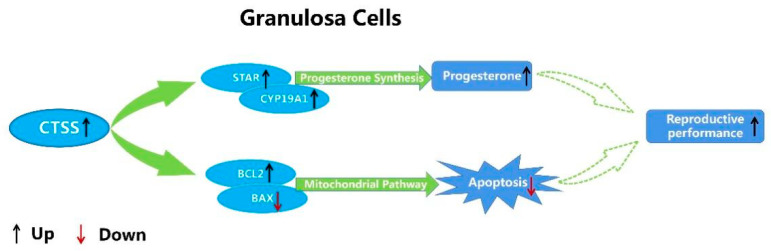
Molecular mechanism of *CTSS* regulating cell apoptosis and progesterone secretion in rabbit GCS.

**Table 1 animals-11-01770-t001:** Name, accession number, sequences, amplicon length of primer pairs of RT-qPCR.

Gene/Accession No.	Primers	Sequence (5′ to 3′)	bp
*PCNA*	F.432	TGCACGTATATGCCGAGACC	240
XM_017341762.1	R.671	GTAGGAGAAAGCGGAGTGGC	
*N* *otch 2*	F.3798	CAACCGCCAGTGTGTTCAAG	232
XM_017345939.1	R.4029	CTTCCGCTTTCGTTTTGCCA	
*Cyclin D*	F.44	GCAGCCCTTTCAATGCTGAC	244
XM_017348091.1	R.697	CTTTGGACGCTCTGACCAGT	
*STAR1*	F.477	GATTGGGAAGGACACGGTCA	179
XM_017350353.1	R.655	CACCCCTGATGACGCCTTT	
*CYP19A1*	F.463	CCTGGGCTTGTTCAGATGGT	171
NM_001170921.2	R.633	ACTTTCGTCCATGGGGATGC	
*CYP11A1*	F.1259	GCCGGGAACCTAGCTTCTTT	155
XM_008253734.2	R.1413	GATGCTCATCTCCACCTCGG	
*BCL2*	F.492	GACTGAGTACCTGAACCGGC	166
XM_008261439.2	R.65	GAGGGTGATGCAAGCTCCTA	
*Bax*	F.11	CCGGGGAGCAGTCCAGA	167
XM_002723696.3	R.177	CAGCTTCTTGGTGGACTCGT	
*Caspase-3*	F.1	ATGGAGAACAACGAAACCTCC	191
NM_001082117.1	R.191	CGGGACGACATTCCAGTGTT	
*β-actin*	F.1140	ATGCAGAAGGAGATCACCGC	148
NM_001101683.1	R.191	ACTCCTGCTTGCTGATCCAC	
*CTSS*	F.128	CTACGGCAAGCAATACAA	187
XM_002715580.3	R.296	TCTCAGGGAACTCATCAAA	

Annealing temperature for all primers listed in this table is 60°C: F, forward primer; R, reverse primer. *PCNA*, proliferating cell nuclear antigen; *STAR1*, steroidogenic acute regulatory protein 1; *CYP19A1*, cytochrome P450 family 19 subfamily A member 1; *CYP11A1*, cytochrome P450 family 11 subfamily 11 subfamily A member 1; *BCL2*, B-cell lymphoma-2.

## Data Availability

Not applicable.

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
