# Peer review of "Modulation of Cathepsin S (CTSS) Regulates the Secretion of Progesterone and Estradiol, Proliferation, and Apoptosis of Ovarian Granulosa Cells in Rabbits"

_animals, 2021, doi:10.3390/ani11061770_

Round 1

Reviewer 1 Report

The manuscript has been improved.

The mention to fig 6 should be integrated into the discussion and not in the conclusion

Author Response

Dear reviewer,

Thank you for giving us the chance to revise our manuscript entitled “Modulation of Cathepsin S (CTSS) Regulates the Secretion of Progesterone and Estradiol, Proliferation, and Apoptosis of Ovarian Granulosa cells in Rabbits” by Guohua Song et al. (manuscript ID: animals-1258103). We are thankful to you for pointing out some important modifications needed in the comment. We have thoughtfully taken into account these comments, the revised parts were highlighted in red in the manuscript.

We believe that the comments have been highly constructive and very useful to restructure the manuscript. We also believe that the latest version really improved the quality of manuscript. We hope that all these changes fulfill the requirements to make the manuscript acceptable for publication in Journal of ANIMALS-BASEL.

Yours sincerely,

GuohuaSong and Huifen Xu on behalf of the authors.

Corresponding author: College of Animal Science and Technology, Henan Agricultural University, Zhengzhou 450046, China, huifen221@126.com.

Point 1: The manuscript has been improved. The mention to fig 6 should be integrated into the discussion and not in the conclusion

Response 1: Thanks for your comments. We have deleted the mention of figure 6 from the conclusion and integrated it into the discussion part as suggested. Please check them in Line 287, 302, 314-315.

Reviewer 2 Report

Dear Authors,

Thank you for addressing my suggestions.

I read your manuscript and your replies. Now I don't have any concerns.

King regards

Author Response

Dear reviewer,

Thanks again for your advice and comments to this manuscript, they are all valuable and very helpful for improving our manuscript before it was published.

Yours sincerely,

GuohuaSong and Huifen Xu on behalf of the authors.

Corresponding author: College of Animal Science and Technology, Henan Agricultural University, Zhengzhou 450046, China, huifen221@126.com.

Reviewer 3 Report

The responses and changes in the manuscript introduced by  authors satisfy and resolve my concerns. Authors' made a good revision work.

Author Response

Dear reviewer,

Thanks again for your advice and comments to this manuscript, they are all valuable and very helpful for improving our manuscript before it was published.

Yours sincerely,

GuohuaSong and Huifen Xu on behalf of the authors.

Corresponding author: College of Animal Science and Technology, Henan Agricultural University, Zhengzhou 450046, China, huifen221@126.com.

This manuscript is a resubmission of an earlier submission. The following is a list of the peer review reports and author responses from that submission.

Round 1

Reviewer 1 Report

The manuscript is in general interesting and well written.

Several improvements are however needed:

Abstract: lines 31-34 - the two latest sentences start with the same words - please rephase

Introduction: 

line 40: a more recent review should be also cited: Applied Sciences | Free Full-Text | Rabbit Genetic Resources Can Provide Several Animal Models to Explain at the Genetic Level the Diversity of Morphological and Physiological Relevant Traits (mdpi.com)

lines 61-62: it is not clear if the location is referred to the rabbit genome or not

lines 72-75: it is not clear what is the real aim of the study 

M&M the number of cell, analyses, tissue specimens per type, should be better defined and clarified

Results: it is not clear where all results related to the expression of several genes reported in Table 1 are reported and if they are relevant or not and in which context - please clarify

Discussion: lines 261-263 - I doubt that this is really something new and relevant

lines 275-278: is there any direct relevant/effect of CTSS or is this related to side or secondary effects ? This could not be clarified by the study

CTSS proteinase activity is well known to be involved in apoptosis processes - therefore the authors should better focus on the novelty of the study 

line 292-293. I do not think this is true/possible - please detete or rephrase

Conclusions should be rewritten - they are almost obvious considering the role of CTSS - in addition is it really new what was reported for the isolation 

of GCS in rabbit. In addition, the final sentence is too bold and does not reflect what is reported by the study

Author Response

Dear reviewer,

Thank you for your comments concerning our manuscript entitled “Cathepsin S (CTSS) Promotes Progesterone secretion and Cell Proliferation in Rabbit Granulosa cells” (animals-1160436) by Guohua Song et al. The comments were all valuable and very helpful for revising and improving our paper. We have studied the comments, read the Instructions for Authors carefully, and revised the manuscript accordingly. The revised portions are highlighted in yellow in the revised manuscript(annex). We hope that all these changes fulfill the requirements to make the manuscript acceptable for publication in Journal of ANIMALS-BASEL.

Looking forward to hearing from you soon.

Yours sincerely,

GuohuaSong and Huifen Xu on behalf of the authors.

Corresponding author: College of Animal Science and Technology, Henan Agricultural University, Zhengzhou 450046, China, huifen221@126.com.

Point 1: Abstract: lines 31-34 - the two latest sentences start with the same words - please rephase

AU: Thank you for your advice. We have revised these sentences and deleted the second “Taken together” in the manuscript as suggested. Please check them in Line 31.

Point 2: line 40: a more recent review should be also cited: Applied Sciences | Free Full-Text | Rabbit Genetic Resources Can Provide Several Animal Models to Explain at the Genetic Level the Diversity of Morphological and Physiological Relevant Traits (mdpi.com)

AU: Thank you for your suggestion. We have cited this paper in the manuscript, see them Line 39 and 315-316.

Point 3: lines 61-62: it is not clear if the location is referred to the rabbit genome or not

AU: We are sorry for the mistake. We have checked the location of rabbits’ CTSS gene in NCBI, it is located on chromosome 13, correction have been made in the manuscript, see them in Line 60-61.

Point 4: lines 72-75: it is not clear what is the real aim of the study 

AU: We appreciate for your valuable comments and we are sorry for the unclear description of our research purpose in the manuscript. This study aimed to investigate the function of CTSS in regulating follicle development and ovulation in rabbits, because cell proliferation, apoptosis and reproductive-related hormone secretion are the main factors that affect follicle development and ovulation, so our main purpose of the study is to investigate the regulatory function of CTSS gene in cell proliferation, apoptosis and hormone (progesterone and estradiol) secretion. To make it more clear, we have rewritten this paragraph in the manuscript, please check them in Line 71-78.

Point 5: M&M the number of cell, analyses, tissue specimens per type, should be better defined and clarified

AU: Thanks for the advice. We have added more information and description in the M&M as suggested. See them in Line 86-89,91-102.

Since the other two reviewers pointed out that tissue expression analysis of CTSS was worthless, and low biological replicates (n=3) used in this experiment will weakens the robustness of the results, so we decided to delete this result, tissue samples collection were also deleted in M&M (Line 87-88).

Point 6: Results: it is not clear where all results related to the expression of several genes reported in Table 1 are reported and if they are relevant or not and in which context - please clarify

AU: Thanks for your question. As a matter of fact, all the genes we measured in the present study (listed in Table 1) are all reported to be involved in cell proliferation, apoptosis and  progesterone, estradiol secretion. PCNA, Notch2, Cyclin D are closely related to cell proliferation. In the study of Ma C et al and Bhagat R et al, the expression of PCNA, Notch2, Cyclin D was used to measure cell proliferation activity (Ma C et al. piRNA-63076 contributes to pulmonary arterial smooth muscle cell proliferation through acyl-CoA dehydrogenase. J Cell Mol Med. 2020 May;24(9):5260-5273; Bhagat R et al. Zika virus E protein alters the properties of human fetal neural stem cells by modulating microRNA circuitry. Cell Death Differ. 2018 Nov;25(10):1837-1854.). STAR1, CYP19A1, CYP11A1 are reported to be responsible for the synthesis of progesterone and estradiol (Strauss J F et al. The steroidogenic acute regulatory protein (StAR): a window into the complexities of intracellular cholesterol trafficking. Recent Prog Horm Res. 1999;54:369-94; discussion 394-5; Xu H et al. Bisphenol A affects estradiol metabolism by targeting CYP1A1 and CYP19A1 in human placental JEG-3 cells. Toxicol In Vitro. 2019 Dec;61:104615). For BCL2, BAX, and CASPASE3, many studies have demonstrated their important role in regulating cell apoptosis, they are always used as marker genes for assess cell apoptosis conditions. A recent study by Liu G et al assessed the apoptosis of ovarian granulosa cells by measuring the expression of these genes by RT-qPCR. (Liu G et al. lncRNA PVT1/MicroRNA-17-5p/PTEN Axis Regulates Secretion of E2 and P4, Proliferation, and Apoptosis of Ovarian Granulosa Cells in PCOS. Mol Ther Nucleic Acids. 2020 Jun 5;20:205-216). Taken together, our present study aimed to investigate the regulatory function of CTSS gene in cell proliferation and apoptosis, estradiol and progesterone secretion in rabbits granulosa cells, so we measured the expression of these listed genes in the present study.

Point 7: Discussion: lines 261-263 - I doubt that this is really something new and relevant

AU: Thanks for your comments. Actually there have been reports about the isolation rabbit granulosa cells. In the present study, the method we used to isolate rabbits granulosa cells are performed according to the published papers of cow, mouse and rabbit with little modifications. So, we have added description of cell isolation and cited the reference of rabbits granulosa cells isolation in the manuscript. See them in Line 260-263. In addition, we also add more details about the isolation and culture of rabbits GCS in the M&M, the published papers we referred is also cited in the M&M, see in Line 91-102.

Point 8: lines 275-278: is there any direct relevant/effect of CTSS or is this related to side or secondary effects ? This could not be clarified by the study

AU: Thanks for your question and comments. We agree with your comments that our results did not provide evidence that CTSS can directly regulate hormone secretion. Although our present study demonstrated that CTSS overexpression and knockdown can increase and decrease the secretion of progesterone and estradiol, this provide direct evidence for a regulatory role of CTSS in the process of progesterone and estradiol secretion. However, hormone secretion is a complex process which is regulated by different factors and pathways, our results did not evidence of CTSS directly regulating hormone secretion and rabbits reproductive performance, the exact mechanism and whether it is a side or secondary effects is not clear, and that’s what we are going to do in our future research. Based on this, we have rewritten this sentence according to your recommendation, see them in Line 272-276.

Point 9: CTSS proteinase activity is well known to be involved in apoptosis processes - therefore the authors should better focus on the novelty of the study     

AU: Thanks for your valuable advice. CTSS is well known for its role in regulating apoptosis processes, the apoptosis of granulosa cells affects follicular development, and excessive apoptosis of granulosa cells will lead to follicular atresia. Rabbits is well known for its high efficiency of reproductive performance, so it is of great importance to investigate the complex mechanism that regulate rabbits reproduction, while there is no report about the regulatory role of CTSS gene in granulosa cells of rabbits. In the present study, we found that CTSS gene is not only involved in cell apoptosis, but also are involved in the process of hormone (progesterone and estradiol) secretion in rabbits granulosa cells, this is the novelty of the study.

Another important reason for the present study is that, we have done some research to investigate the effects and its biological significance of fasting caecotrophy on rabbits growth, development and reproduction, our results found that fasting caecotrophy will decrease the growth and development of rabbits, and also resulted in fetal absorption of pregnancy rabbits. We submitted the oocytes of rabbits in the control group and fasting caecotrophy group for RNA-sequencing, results indicated that CTSS is markedly decreased by fasting caecotrophy, so we speculated that CTSS plays an important role in the process of caecotrophy-mediated reproduction failure. Thus, the purpose of the present study is to validate the regulatory role of CTSS in rabbits granulosa cells. However, these data are only partly published (Ruiting Li et al., Influence of cecotrophy on fat metabolism mediated by caecal microorganisms in new zealand white rabbits, J Anim Physiol Anim Nutr, 2020, 104(2): 749-757; Yadong Wang et al., Transcriptome analysis of the effects of fasting caecotrophy on hepatic lipid metabolism in new zealand rabbits, Animals, 2019, 9(9): 648), the reproduction data have not been published, so we did not mention this novelty of the purpose in the manuscript.

Point 10: line 292-293. I do not think this is true/possible - please detete or rephrase

AU: Thanks for your advice. We have deleted this sentence in the manuscript as suggested.

Point 11: Conclusions should be rewritten - they are almost obvious considering the role of CTSS - in addition is it really new what was reported for the isolation of GCS in rabbit. In addition, the final sentence is too bold and does not reflect what is reported by the study

AU: Thanks for your valuable advice. Actually, the method we used to isolate rabbits GCS in the present study is not new, we isolated rabbits GCS according to the published papers with little modifications, we have revised the description of rabbit GCS isolation in the manuscript as suggested (Point 7 also emphasized this issue, see them in Line 260-263). The conclusions of the manuscript have been rewritten and the final sentence have been deleted as suggested, please see them in Line 291-298.

Reviewer 2 Report

Dear Authors,

I apologize but I consider the publication of the manuscript in the present form to be inappropriate.

There are too many concerns in the experimental design, methodology and statistical analysis.

It is hard to follow the manuscript and the title is not coherent with the text

There are many acronyms without any definition the first time they appear, for example:

Line 87 PBS

Line 96 FSHR

Line 105 HE

Line 111 siRNA

Line 179 DAPI

Line 71 remove the comma and insert the full stop.

There are many mistakes in the table 1 where the primers are listed. Some primers and acc. number are incorrect and this issue doesn’t allow to repeat the experiment. Some examples:

In NM_001170921.2 the primer F 463 doesn’t align with the sequence

NM_001101683.1 is not the sequence of Capsase-3

NM_001082117.1 is not the sequence of b-actin

The statistical analysis is too poor.

The only information on the statistic procedures are the one-way ANOVA to compare the effects of treatments. What are the treatments?  The statistical analysis is too approximate. It seems that the only aim of the study was to compare the efficiency of CTSS overexpression and siRNA interference

Nothing about “relative mRNA expression of CTSS in different tissues” (line 169 and figure 1)

Nothing about the aim of the research (line 72-73 “In this study, we attempted to measure the effects of CTSS on the proliferation and apoptosis, as well as progesterone and estradiol secretion of GCS”)

Author Response

Dear reviewer,

Thank you for your comments concerning our manuscript entitled “Cathepsin S (CTSS) Promotes Progesterone secretion and Cell Proliferation in Rabbit Granulosa cells” (animals-1160436) by Guohua Song et al. The comments were all valuable and very helpful for revising and improving our paper. We have studied the comments, read the Instructions for Authors carefully, and revised the manuscript accordingly. The revised portions are highlighted in yellow in the revised manuscript(annex). We hope that all these changes fulfill the requirements to make the manuscript acceptable for publication in Journal of ANIMALS-BASEL.

Looking forward to hearing from you soon.

Yours sincerely,

GuohuaSong and Huifen Xu on behalf of the authors.

Corresponding author: College of Animal Science and Technology, Henan Agricultural University, Zhengzhou 450046, China, huifen221@126.com.

Point 1: It is hard to follow the manuscript and the title is not coherent with the text

AU: We really appreciate your comments. We are sorry for the unclear description of the manuscript, we have revised the manuscript according to the reviewers’ suggestions and sent this manuscript to MDPI for language improving, we believe that will improve the language and quality of the manuscript.

For the title of the text, this paper aimed to investigate the function of CTSS in regulating follicle development and ovulation in rabbits, while cell proliferation, apoptosis and reproductive-related hormone secretion are the main factors that affect follicle development and ovulation, and our results indicated that CTSS can promote cell proliferation and hormone (progesterone and estradiol) secretion. In this paper, we did not get direct evidence showing that CTSS can ultimately affect follicle development and ovulation through regulating cell proliferation and hormone secretion. However, we agree with you that the title is not proper, so we have changed the tile into “Modulation of Cathepsin S (CTSS) Regulates the Secretion of Progesterone and Estradiol, Proliferation, and Apoptosis of Ovarian Granulosa cells in Rabbits”, please see them in the manuscript.

Point 2: There are many acronyms without any definition the first time they appear, for example:

Line 87 PBS 

Line 96 FSHR 

Line 105 HE 

Line 111 siRNA 

Line 179 DAPI  

Line 71 remove the comma and insert the full stop.

AU: We are sorry for the lack of definition. We have checked the manuscript and added the definition according to the suggestions, please check them in Line 93 (PBS), 104 (FSHR), 110-111 (DAPI), 113 (HE), 120 (siRNA) and 134-135 (MOI). The comma was deleted and inserted with the full stop as suggested (Line 70).

Point 3: There are many mistakes in the table 1 where the primers are listed. Some primers and acc. number are incorrect and this issue doesn’t allow to repeat the experiment. Some examples:

In NM_001170921.2 the primer F 463 doesn’t align with the sequence

NM_001101683.1 is not the sequence of Capsase-3

NM_001082117.1 is not the sequence of b-actin

AU: Thanks for your comments, we are very sorry for the mistakes. We have checked all the primer sequences and acc. number in table 1, and have revised the mistakes in the manuscript, see them in table 1.

Point 4: The statistical analysis is too poor.

AU: Thanks, we agree with you that the statistical analysis it too poor. We have added more description about the detail of statistical analysis, see them in Line 172-178.

Point 5: The only information on the statistic procedures are the one-way ANOVA to compare the effects of treatments. What are the treatments?  The statistical analysis is too approximate. It seems that the only aim of the study was to compare the efficiency of CTSS overexpression and siRNA interference.

AU: We appreciate for your question and valuable comments. We have added more description of details in the statistics analysis, please see them in Line 172-178. And, to be specific, the aim of this study was to investigate the function of CTSS in regulating follicle development and ovulation in rabbits. While cell proliferation, apoptosis and reproductive-related hormone secretion are the main factors that affect follicle development and ovulation, so our main purpose of the study is to investigate the regulatory function of CTSS gene in cell proliferation, apoptosis and hormone (progesterone and estradiol) secretion. In the present study, we speculated that manipulation of CTSS gene might bring about changes in cell proliferation and hormone secretion. In order to testify this hypothesis, gene overexpression  and knockdown are used to investigate the effects of CTSS gene on cell proliferation and hormone secretion. The aim for measuring the efficiency of CTSS overexpression and siRNA interference is to make sure CTSS is successfully overexpressed and knockdown, so that they can be used in further investigating the function of CTSS in cell proliferation and hormone secretion in rabbits GCS.

Point 6: Nothing about “relative mRNA expression of CTSS in different tissues” (line 169 and figure 1)

AU: Thanks for your comments. We agree with you that the tissue expression profile of CTSS did not bring about much information for the manuscript, and low biological replicates (n=3) used in this experiment will weakens the robustness of the results, so we decided to delete this result. Descriptions regarding the tissue expression profile of CTSS are all deleted in the manuscript, please check them in the manuscript.

Point 7: Nothing about the aim of the research (line 72-73 “In this study, we attempted to measure the effects of CTSS on the proliferation and apoptosis, as well as progesterone and estradiol secretion of GCS”)

AU: Thanks for your comments. We are sorry for the unclear description about the research aim of the manuscript, this issue is also raised by another reviewer. To make it more clear, we have rewritten that paragraph in the manuscript. As shown in Line 71-78.

Reviewer 3 Report

The manuscript of Guohua Song et al. with the title “Cathepsin S (CTSS) Promotes Progesterone secretion and Cell 2 Proliferation in Rabbit Granulosa cells” aims to study the role of Cathepsin S in regulating cell apoptosis and progesterone and estrogen secretion in rabbit granulosa cells. The subject under investigation is of interest in an area still little explored from the scientific point of view with potential impact on manipulation of rabbit’s reproduction. The article is well written, and the experimental design is adequate to the defined goals.

Major concern:

Although this work can be considered exploratory in nature, the number of individuals used is very low (n = 3), which weakens the robustness of the results.

Minor concerns:

Line 46 - Granular cells – did you mean granulosa cells or follicular cells

In the abstract are claimed that “Tissue expression analysis revealed a high 23 expression of CTSS in spleen, liver, heart, lung and ovary”, however in results (line 171) authors missed lung “Spleen got the highest expression of CTSS, followed by heart and liver, ovary tissue…”

On conclusions authors claim that “CTSS plays an important role in the regulation of rabbit reproduction”. Please change for the form of suggestion (speculative).

Author Response

Dear reviewer,

Thank you for your comments concerning our manuscript entitled “Cathepsin S (CTSS) Promotes Progesterone secretion and Cell Proliferation in Rabbit Granulosa cells” (animals-1160436) by Guohua Song et al. The comments were all valuable and very helpful for revising and improving our paper. We have studied the comments, read the Instructions for Authors carefully, and revised the manuscript accordingly. The revised portions are highlighted in yellow in the revised manuscript(annex). We hope that all these changes fulfill the requirements to make the manuscript acceptable for publication in Journal of ANIMALS-BASEL.

Looking forward to hearing from you soon.

Yours sincerely,

GuohuaSong and Huifen Xu on behalf of the authors.

Corresponding author: College of Animal Science and Technology, Henan Agricultural University, Zhengzhou 450046, China, huifen221@126.com.

Point 1: Although this work can be considered exploratory in nature, the number of individuals used is very low (n = 3), which weakens the robustness of the results.

AU: Thanks for your valuable comments. We agree with the reviewer’s comments that 3 biological replicates is not enough for in vivo experiments, and another reviewer also pointed out that the result of tissue expression profile of CTSS did not bring about much information, so we decided to delete this experiment and results from the present manuscript. Descriptions regarding the tissue expression profile of CTSS are all deleted in the manuscript, please check them in the manuscript.

However, we really appreciate for your comments and we agreed with your opinion, and we will use 5-6 biological replicates in our following research.

Point 2: Line 46 - Granular cells – did you mean granulosa cells or follicular cells

AU: We are sorry for the unclear description. In the present study, “Granular cells” means “granulosa cells”. Recent studies always use the term “granulosa cells” (Liu G et al., lncRNA PVT1/MicroRNA-17-5p/PTEN Axis Regulates Secretion of E2 and P4, Proliferation, and Apoptosis of Ovarian Granulosa Cells in PCOS. Mol Ther Nucleic Acids. 2020 Jun 5;20:205-216). To be more specific, we have changed “granular cells” into “granulosa cells” in the manuscript. Please check them in the manuscript.

Point 3: In the abstract are claimed that “Tissue expression analysis revealed a high 23 expression of CTSS in spleen, liver, heart, lung and ovary”, however in results (line 171) authors missed lung “Spleen got the highest expression of CTSS, followed by heart and liver, ovary tissue…”

AU: Sorry for the mistake. Since that we have deleted the portions regarding tissue expression profile of CTSS according to reviewers’ comments, so these two sentences have been deleted too in the manuscript, please check them in the manuscript.

Point 4: On conclusions authors claim that “CTSS plays an important role in the regulation of rabbit reproduction”. Please change for the form of suggestion (speculative).

AU: We appreciate your comments and suggestion. Conclusions have been rewritten, please see them in Line 291-298.

We appreciate your earnest work, and hope that the corrections will meet their approval. Once again, thank you very much for your comments and suggestions.
